# The Effect of an Abnormal BMI on Orthopaedic Trauma Patients: A Systematic Review and Meta-Analysis

**DOI:** 10.3390/jcm9051302

**Published:** 2020-05-01

**Authors:** Florence Kinder, Peter V. Giannoudis, Tim Boddice, Anthony Howard

**Affiliations:** 1Undergraduate Department, School of Medicine, Leeds University, Leeds LS2 9JT, UK; um17frk@leeds.ac.uk; 2Leeds Orthopaedic Trauma Sciences, LGI, University of Leeds, Leeds LS2 9JT, UK; P.Giannoudis@leeds.ac.uk (P.V.G.); timothy.boddice@gmail.com (T.B.); 3Academic Department of Trauma and Orthopaedics, LGI, University of Leeds, Leeds LS2 9JT, UK; 4NIHR Leeds Biomedical Research Centre, Chapel Allerton Hospital, Leeds LS7 4SA, UK

**Keywords:** BMI, surgical complications

## Abstract

Aims: The aim of this systemic review is to identify the complications that arise in operating on orthopaedic trauma patients with an abnormal body mass index (BMI). Materials and Methods: Systematic literature search using a combination of MESH subject headings and free text searching of Medline, Embase, SCOPUS and Cochrane databases in August 2019. Any orthopaedic injury requiring surgery was included. Papers were reviewed and quality assessed by two independent reviewers to select for inclusion. Where sufficiently homogenous, meta-analysis was performed. Results: A total of 26 articles (379,333 patients) were selected for inclusion. All complications were more common in those with a high BMI (>30). The odds ratio (OR) for high BMI patients sustaining post-operative complication of any type was 2.32 with a 23% overall complication rate in the BMI > 30 group, vs. 14% in the normal BMI group (*p* < 0.05). The OR for mortality was 3.5. The OR for infection was 2.28. The OR for non-union in tibial fractures was 2.57. Thrombotic events were also more likely in the obese group. Low BMI (<18.5) was associated with a higher risk of cardiac complications than either those with a normal or high BMI (OR 1.56). Conclusion: Almost all complications are more common in trauma patients with a raised BMI. This should be made clear during the consent process, and strategies developed to reduce these risks where possible. Unlike in elective surgery, BMI is a non-modifiable risk factor in the trauma context, but an awareness of the complications should inform clinicians and patients alike. Underweight patients have a higher risk of developing cardiac complications than either high or normal BMI patient groups, but as few studies exist, further research into this group is recommended.

## 1. Introduction 

The impact of obesity and increased body mass index (BMI) upon health services is a growing concern both nationally and internationally [1]. Elective orthopaedic surgery affords the opportunity for the balance of risk and adequate preparation of the patient; however, this is rarely the case in trauma surgery. 

Orthopaedic injury is common in major trauma, with 78% of severely injured patients having significant fracture related injury [2]. A survey in 2013 by Scholes et al. demonstrated that up to 44% of adults in the UK will sustain a fracture in their lifetime. The prevalence of obesity is increasing globally [1], with 13% of the world’s population now classed as obese, a threefold increase over the last 40 years [3]. These trends mean that the obese patient with significant orthopaedic trauma requiring surgery is now a regular presentation in orthopaedic trauma practice. 

The definitions of abnormal BMI remain controversial. BMI is calculated by dividing weight (kilograms) by height squared (metres). This makes it an imprecise measure of obesity, especially in those with a high muscle mass or at extremes of height. However, it is commonly used in both the literature and clinical practice as it is simple and reproducible to measure. For the purpose of the analysis, an abnormal BMI was adopted as below 18.5 and above 30, as defined by the World Health Organisation [4], due to the manner of reporting, the categories of normal and overweight have been amalgamated where necessary. Whilst an exact BMI is unlikely to be measured in an acute trauma setting, owing to difficulties obtaining an accurate weight in an immobile patient, it remains a simple and commonly used measure, using a combination of estimation and patient reported figures. 

An abnormal BMI in either direction has a negative impact on almost all health outcomes [1,5], and obesity as a risk factor increases mortality by a factor of almost two [6]. Multiple studies have also shown obesity as an independent risk factor for mortality following high energy, blunt force trauma [7,8] with increased risks of perioperative and post-operative complications [5]. These patients present unique pathophysiological and physical challenges, ranging from the use of specialised operating tables and equipment [6] to post-operative outcomes and late complications. The risk is not confined to obese patients; underweight patients also show increased perioperative mortality in trauma surgery [9]. Problems with each body system, including the cardiovascular, respiratory, endocrine, gastrointestinal, musculoskeletal and renal systems, have also been reported with increased BMI. Parratte et al., 2014 reported that obese patients are more likely to present with cardiovascular disease, type 2 diabetes and osteoarthritis [10]. Furthermore, airway management can be more difficult with an increased risk of postoperative pulmonary complications, owing to a higher incidence of atelectasis and impairment of functional residual capacity [11]. Poor rates of wound healing may also be seen in patients with a high BMI [12,13]. Notably, mental health conditions such as depression are associated with patients with an abnormal BMI, and evidence points towards the negative implications this may have on recovery, such as prolonged length of stay [14,15]. On the other hand, the ‘obesity paradox’, where obesity is postulated to provide some level of protection in orthopaedic injuries, was referenced by several studies [6,16,17]. Notably, a meta-analysis [18] postulated that a higher BMI may provide protection for patients with hip fractures, due to the excess adipose tissue providing cushioning for the femur and pelvis but also the higher regional bone mineral density strengthening the fracture. 

Consequently, the aim of this systemic review was to identify complications that arise in operating on orthopaedic trauma patients with an abnormal BMI. Understanding the increased risks these patients face is critical to ensure that patients are optimised and appropriately counselled of the additional risks and precautions associated with body habitus. Similarly, orthopaedic surgeons and all who care for such patients should be aware of these complications in order to employ appropriate measures to ensure the best possible outcomes. 

## 2. Methods 

The following electronic bibliographic databases were used: Ovid Medline In-Process and Other Non-Indexed Citations, Ovid MEDLINE, Ovid EMBASE, Ovid Cochrane, Scopus (Supplementary materials). A search including all eligible studies from the decade preceding 1 August 2019 was conducted. The topic keywords entered were BMI, Body Mass Index, Surgery, Orthopaedic and Trauma, and 28 associated MESH terms were used to further expand the search. The search was limited to the English language and human subjects. Controlled vocabulary based on the assessment of the patient, intervention, comparison and outcome (PICO) was used to search for studies involving adult acute orthopaedic surgery, comparing normal BMI patients to abnormal. Abnormal BMI was defined as <18.5 and >30. Case reports and case control studies were excluded. A rigorous systematic approach, adopting the method set out in the Cochrane [19] handbook for systematic reviews of interventions was carried out. The review was registered on the PROSPERO database on 12 June 2019 (No. 138813). No funding was received for the review.

An independent review by two investigators (Anthony Howard and Florence Kinder) was undertaken initially of the literature. The Preferred Reporting Item for Systematic reviews and Meta-Analysis (PRISMA statement) was used to report the findings. Further, searches were undertaken to identify references contained in bibliographies and other specified orthopaedic injuries or conditions. 

### Statistical Analysis 

Data were extracted with regard to complications associated with orthopaedictrauma surgery. The odds ratio was the primary measure of the overall complication and mortality. Study and participant characteristics were recorded (year, BMI, complications). Bias was assessed using relevant parts of the Newcastle-Ottawa Quality Assessment Scale: representativeness of the exposed cohort, ascertaining of exposure, assessment of outcome, and adequacy of follow-up of cohorts and GRADE [20,21,22]. Both investigators undertook the assessment independently.

## 3. Results

The initial search revealed 1586 articles, out of which 239 were selected by two investigators (AH and FK) for full text review following review of the title and abstract (Figure 1). The 239 studies which underwent full text review resulted in a further 213 studies being excluded, primarily owing to either ambiguous definitions of abnormal BMI or non-complication related data. 

Twenty-six articles, involving a total of 379,333 patients, were ultimately included in the study (Table 1). There were two papers [23,24] recording complications in patients with low BMI (<18.5), involving 416 patients. Due to the manner of reporting, complications were often grouped together despite being heterogenous in nature and onset from the index accident. In the 26 papers, the following grouping could be established; overall complications (*n* = 13), mortality (*n* = 7), infection (*n* = 15), cardiac event (*n* = 5), deep vein thrombosis (*n* = 7) and renal failure (*n* = 1).

In papers (*n* = 9) [23,24,26,27,28,35,38,41,43] recording the odds ratio, patients with a high BMI (>30) had an overall complication odds ratio of 2.32 (Figure 2). In patients with a BMI < 18.5, the odds ratio of complications was 1.36.

Overall complications odds ratios (OR) were available for 13,868 obese patients [23,24,26,27,28,35,38,41,43] and 416 underweight patients [23,24] compared to patients with a normal BMI. On meta-analysis, OR for overall complications were calculated at 2.32 and 1.36, respectively (Figure 3). Raw data on complications were available for 21,868 patients (2572 obese, 19,296 normal BMI) [29,32,40]. Overall complications were seen in 14.7% of patients with a normal BMI and 23.9% with a BMI over 30 (*p* < 0.05). Mortality rates in 824 patients [23] (184 underweight, 640 normal BMI) were reported at 9.3% and 4.4%, respectively. In 1137 patients [29,36,38] (651 obese, 486 normal), mortality rates were reported at 4.9% in the obese, compared to an average 2.4% in patients with a BMI under 30. OR for mortality in 678 obese patients [35] was reported at 3.5. 

Infections and wound problems in 30,176 normal patients [27,29,30,31,36,40,41,43,45] were averaged at 5.6% compared to an average of 12.7% in 5987 obese patients [27,29,30,31,36,40,43,45] *p* < 0.05. Different types of fixation [27,45] were averaged for the purpose of general analysis. An average OR of 2.28 was reported in 6334 obese patients [26,27,31,33,37,39,45] (Figure 4). For the purpose of this analysis categorised OR were averaged. This included 164 obese patients who suffered superficial (1.79) and deep infections (2.09) [39]. A total of 2841 underwent open reduction and internal fixation (ORIF) (1.9 (27), 3.2 (46)), with 967 undergoing intramedullary nailing (IMN) (1.4 [27], 3.4 [45]), 821 undergoing hemiarthroplasty for proximal humeral shaft fractures (4 [45]), and 256 undergoing total shoulder arthroplasty for proximal humeral shaft fractures (2.3 [45]). Data were not found for underweight patients. 

Obese patients were shown to have longer hospital stays, on average an additional 1.3 days in 9916 obese patients [28,29,32,38]. Data were not found for underweight patients. 

Revision surgery rates were recorded in 723 patients [21] (564 normal and 159 obese), the obese had a hazard ratio of 3.08 and 3.10 for metal work failure. Rates of metal work failure and non-union were recorded in 13,818 patients (12,727 normal, 1091 obese) [26], and of these an average of 7.7% of normal BMI patients had metal work failure compared to 8.2% obese. Non-union was reported in 11.1% of normal and 16.2% of obese (Figure 5). The OR for non-union in 50 obese patients [37] with tibial shaft fractures was 2.57. 

In respect to cardiac and deep vein thrombosis (DVT), the OR of cardiac complications was reported for 2360 obese patients, and the average OR was calculated at 1.94 [21,25,26]. Furthermore, the proportion of patients experiencing cardiac complications was considered in 379 normal patients and 530 obese patients; on average, 2.25% of patients with a normal BMI experienced cardiac complications (including arrhythmias), compared with 3.65% of obese patients [36,38]. The risk of any event is higher in the underweight OR (1.56) when compared with normal BMI and obese. 

Incidence of DVT was assessed in 1763 patients (1202 normal BMI, 561 obese) [20,23,29,38,40,41]. In total, 1.28% of normal BMI patients reported DVT, compared with an average of 3.18% of obese. 

## 4. Discussion 

As the global epidemic of obesity and the incidence of eating disorders continues to rise, the number of patients with abnormal BMI and significant orthopaedic trauma will also rise.

Identification of specific risk is important, and should guide individual surgeons and units to develop strategies to mitigate these enhanced risks. 

An increase in BMI brings about pathophysiological changes in almost all organ systems, including cardio-vascular disease, diabetes mellitus and cancer, largely as a display of ‘metabolic syndrome’ [46]. Positive energy-balance, with more calories consumed than expended and consideration of adipose distribution, with particular concern given to intrabdominal accumulation, can explain such change. Excess accumulation around structures can result in compression leading to complications such as hypertension with renal compression and sleep apnea as a result of pharyngeal compression, both of which hold potential for anaesthetic complications [47]. 

Production of proinflammatory adipokines by excess adipocytes can result in persistent systemic inflammation in some obese patients which may have implications on wound healing and infection [49]. 

Excess weight puts increased pressure on joints and is itself a risk factor for osteoarthritis, particularly of the knee joint. OA has impacts on mobility and rehabilitation, both key considerations following orthopaedic trauma [48]. Alongside physical changes, it is vital we consider and recognise the psychiatric implications of obesity and the pathophysiological consequences resulting in increased prevalence of depression and other mental health conditions. The exact physiology behind this is unknown but suggestions such as increased social isolation due to societal stigma is proposed [48,50]. 

Patients with an abnormal BMI present many challenges for orthopaedic surgeons at all stages of their care. Pre-operatively, the challenge of stabilisation, access to adequate imaging and sourcing of specialist implants and operative equipment, e.g., bariatric operating tables, can delay surgery. Furthermore, excess skin can make access difficult and can create difficulties surrounding sterility if large areas are exposed. During operation, as discussed previously, anaesthesia and airway management can be challenging both from an intubation/bag-mask ventilation, extubation and drug perspective, due to ‘poor respiratory mechanics’ and less oxygen reserve and altered dosing requirements. Alongside increased levels of intraoperative blood loss, patients generally have longer operative times, which can lead to increased risk of nerve palsy [51]. 

Post-operatively, as the review indicates, patients with an abnormal BMI face higher rates of complications, from infection to poor mobilisation [52].

This review indicates that compared to patients with a normal BMI, obese orthopaedic trauma patients face greater risk of overall complications, including mortality, wound problems, cardiac events and thrombotic events. They are also more likely to need reintervention for metal work failure or non-union. Metal work failure is higher in patients who undergo IMN compared with ORIF, whilst non-union is seen more frequently in ORIF. Table 2 indicates the prevalence of these complications, with non-union and infection/wound problems being highest, and DVT being the lowest. Furthermore, obese patients have worse functional outcomes and are less likely to achieve complete bone union. This is generally reflective of the study by Chesser et al. in 2010. There is no direct reflection of the mentioned ‘obesity-paradox’ when compared to a normal BMI.

An increased complication rate has led to an increase in the average length of stay, and so an increased cost of care per episode. With a mean cost per patient/day of GBP 143.20 on a standard orthopaedic ward and staffing costs of GBP 155.46, ward-based care alone could add GBP 388.26 to an inpatient stay. This cost is, without factoring in the much higher cost of high dependant unit (HDU)/critical care unit(CCU) care, GBP 559.42/day, which is likely to be required by patients with major complications [53,54,55,56].

The Forrest plot indicates that underweight trauma patients also face worse outcomes, with increased overall complications and a greater mortality incidence compared to normal patients. This is comparable with Whiting et al. [9]. Surprisingly, underweight patients have increased rates of cardiac complications compared to all groups, despite it being thought that obese patients are more likely to present with cardiovascular disease [10,23].

## 5. Conclusion

As a high BMI increases the risk of complications related to post orthopaedic trauma surgery, consideration of this should be at the forefront when faced with obese trauma patients and, where possible, strategies should be created to mitigate the additional risks and costs. In this patient group, the index of suspicion of infection/non-union should be higher, with the threshold to be lower for starting antibiotics and other interventions where appropriate. Post-operatively, nutritional advice should be sought to address deficiencies in macro and micro dietary requirements, along with lifestyle support. There is an urgent need for further research into strategic interventions in this patient group to improve outcomes and reduce complications.

Finally, a low BMI appears to increase the risk of mortality and cardiac complications but limited data for this population restrict any firm conclusions. Further research into this group is recommended.

## Figures and Tables

**Figure 1 jcm-09-01302-f001:**
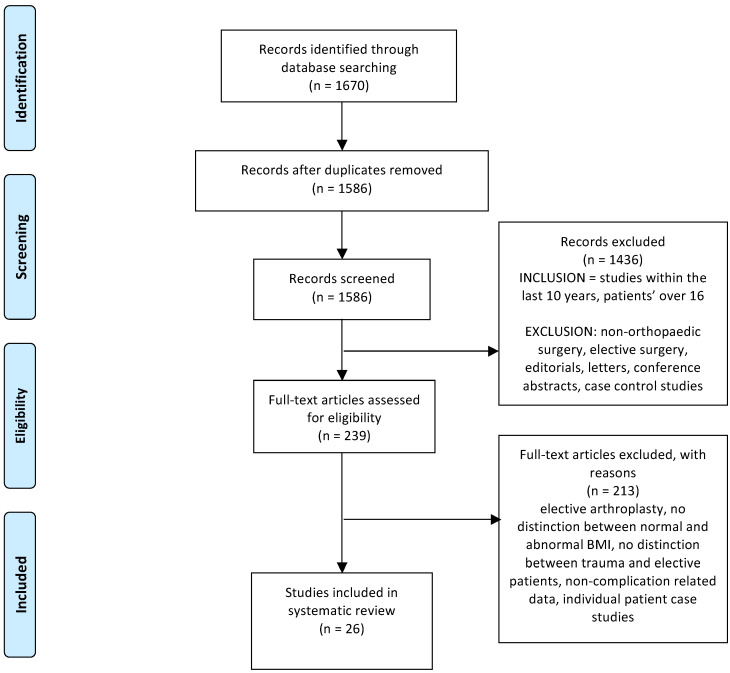
A literature search of abnormal body mass index (BMI), PRISMA, Preferred Reporting. Items for Systematic Reviews and Meta-Analysis.

**Figure 2 jcm-09-01302-f002:**
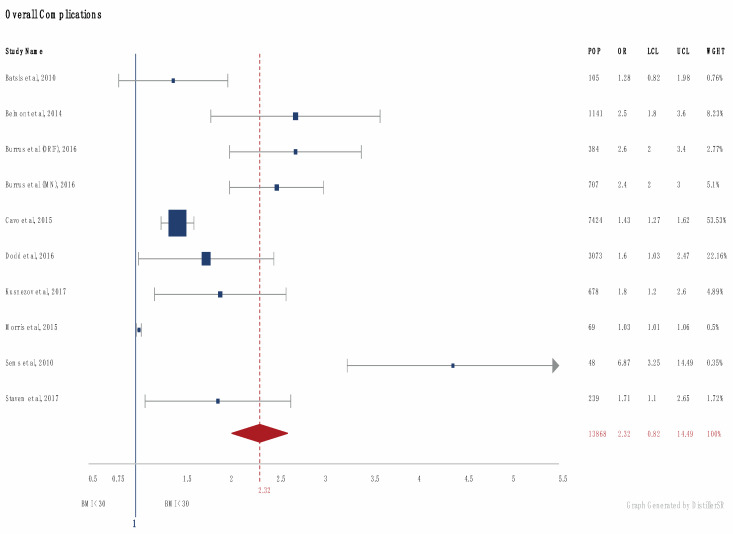
Forrest plot of overall complications in patients >30 Body Mass Index (BMI). (Number of patients (POP), Odds Ratio (OR), Upper (UCL) and Lower (LCL) Confidence interval and Weight (WGHT). Square reflects the number of patients and the diamond represents the total effect.

**Figure 3 jcm-09-01302-f003:**
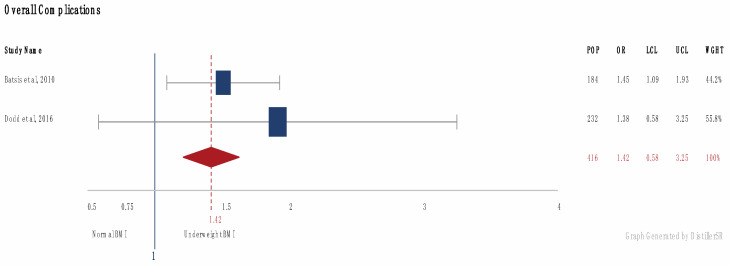
Forrest plot of overall complications in patients < 18.5 BMI). (Number of patients (POP), Odds Ratio (OR), Upper (UCL) and Lower (LCL) Confidence interval and Weight (WGHT). Square reflects the number of patients and the diamond represents the total effect.

**Figure 4 jcm-09-01302-f004:**
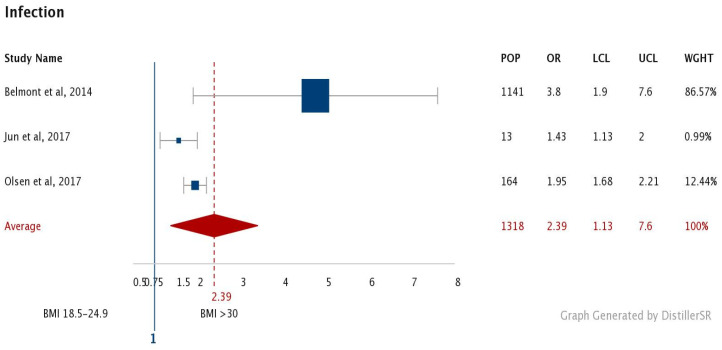
Forest plot comparing Odds Ratio (OR) for infections in the obese. (Number of patients (POP), Odds Ratio (OR), Upper (UCL) and Lower (LCL) Confidence interval and Weight (WGHT). Square reflects the number of patients and the diamond represents the total effect.

**Figure 5 jcm-09-01302-f005:**
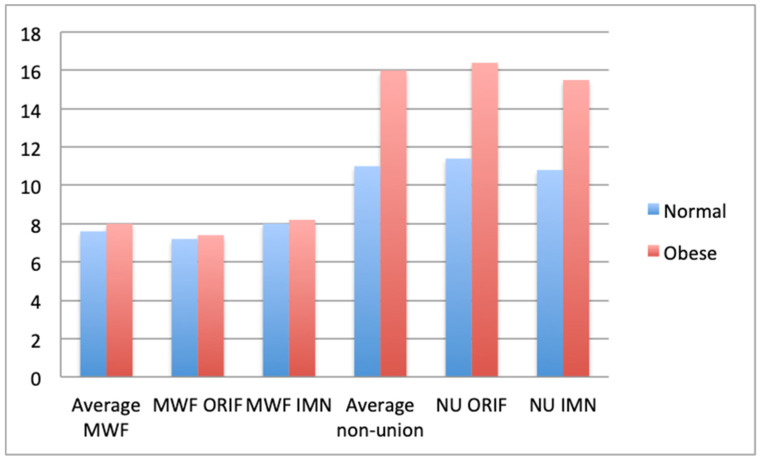
Comparison of % of normal and obese patients requiring reintervention for metal work failure or non-union, averages and surgical specific (MWF = metal work failure; NU= non-union; ORIF = open reduction and internal fixation; IMN = intramedullary nailing).

**Table 1 jcm-09-01302-t001:** Outline of article included in the review and respective patient populations.

ID	Author	Title	Year	Normal BMI	Under weight BMI	Obese BMI	GRADE
1	Ahmad et al. [23]	The Effect of Obesity on Surgical Treatment of Achilles Tendon Ruptures	2017	44	0	32	Low
2	Arroyo et al. [25]	Pelvic trauma: What are the predictors of mortality and cardiac, venous thrombo-embolic and infectious complications following injury?	2013	40,183	0	1114	Mod
3	Baldwin et al. [26]	Does morbid obesity negatively affect the hospital course of patients undergoing treatment of closed, lower-extremity diaphyseal long-bone fractures?	2011	19,795	0	331	Low
4	Batsis et al. [27]	Body mass index (BMI) and risk of noncardiac postoperative medical complications in elderly hip fracture patients: a population-based study	2010	640	184	105	Low
5	Batsis et al. [28]	Body mass index and risk of adverse cardiac events in elderly patients with hip fracture: a population-based study	2009	640	184	105	Low
6	Belmont et al. [29]	Risk factors for complications and in-hospital mortality following hip fractures: a study using the National Trauma Data Bank	2014	43,278	0	1141	Low
7	Burrus et al. [30]	Obesity is associated with increased postoperative complications after operative management of tibial shaft fractures	2016	12,727	0	1091	Mod
8	Cavo et al. [24]	Association Between Diabetes, Obesity, and Short-Term Outcomes Among Patients Surgically Treated for Ankle Fracture	2015	116,411	0	7424	High
9	Childs et al. [31]	Obesity Is Associated With More Complications and Longer Hospital Stays After Orthopaedic Trauma	2015	107	0	121	Low
10	Dincel et al. [32]	Effect of BMI on outcomes of surgical treatment for tibial plateau fractures: A comparative retrospective case study	2018	22	0	16	Very Low
11	Dodd et al. [33]	Predictors of Adverse Events for Ankle Fractures: An Analysis of 6800 Patients	2016	1242	232	3073	High
12	Graves et al. [34]	Is obesity protective against wound healing complications in pilon surgery? Soft tissue envelope and pilon fractures in the obese.	2010	83	0	31	Low
13	Johnson et al. [35]	Increased incidence of vascular injury in obese patients with knee dislocations.	2018	19,087	0	2265	Mod
14	Jun et al. [36]	Risk factors of wound infection after open reduction and internal fixation of calcaneal fractures.	2017	286	0	13	Very Low
15	Koerner et al. [37]	Femoral malrotation after intramedullary nailing in obese versus non-obese patients.	2014	111	0	95	Low
16	Kusnezov et al. [38]	Predictors of inpatient mortality and systemic complications in acetabular fractures requiring operative treatment.	2017	6763	0	678	Mod
17	Maheswari et al. [39]	Severity of injury and outcomes among obese trauma patients with fractures of the femur and tibia: A crash injury research and engineering network study.	2009	204	0	461	Low
18	Metes makers et al. [40]	Individual risk factors for deep infection and compromised fracture healing after intramedullary nailing of tibial shaft fractures: a single centre experience of 480 patients.	2015	430	0	50	Very Low
19	Morris et al. [41]	Obesity Increases Early Complications After High-Energy Pelvic and Acetabular Fractures	2015	175	0	69	Low
20	Olsen et al. [42]	The impact of lifestyle risk factors on the rate of infection after surgery for a fracture of the ankle	2017	879	0	164	Mod
21	Porter et al. [43]	Operative experience of pelvic fractures in the obese.	2008	102	0	186	Mod
22	Sems et al. [44]	Elevated body mass index increases early complications of surgical treatment of pelvic ring injuries.	2010	134	0	48	Low
23	Shubiya et al. [45]	Incidence of Acute Deep Vein Thrombosis and Pulmonary Embolism in Foot and Ankle Trauma: Analysis of the National Trauma Data Bank	2012	72,896	0	2768	High
24	Stavem et al. [46]	The association of body mass index with complications and functional outcomes after surgery for closed ankle fractures.	2017	272	0	239	Mod
25	Weinlein et al. [47]	Morbid obesity increases the risk of systemic complications in patients with femoral shaft fractures.	2015	184		114	Mod
26	Werner et al. [48]	Obesity is associated with increased postoperative complications after operative management of proximal humerus fractures.	2015	16525	0	3794	High

**Table 2 jcm-09-01302-t002:** Order of prevalence of complications (Body Mass Index (BMI)Deep Vain Thrombosis(DVT)).

Complication	Prevalence in BMI <30 (Normal)	Prevalence in BMI >30 (abnormal)
Non-Union	11.1%	16.2%
Infection/Wound Problems	5.6%	12.7%
Metalwork Failure	7.7%	8.2%
Mortality	3.4%	4.9%
Cardiac	2.3%	3.7%
DVT	1.3%	3.2%

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
