# Peer review of "The Effect of an Abnormal BMI on Orthopaedic Trauma Patients: A Systematic Review and Meta-Analysis"

_jcm, 2020, doi:10.3390/jcm9051302_

Round 1

Reviewer 1 Report

Comments and Suggestions for Authors

Obesity is certainly a challenge and as you point out not something that can be changed in an acute trauma situation.  It is difficult to know whether nonunion and failure of fixation was related to infection or metabolic (poor nutrition, low Vitamin D) reasons. I think providing suggestions in the Conclusion for the most significant findings would be important. If infection is the most significant complication is there any evidence that how we prep or redosing antibiotics for longer duration surgery or continuation of antibiotics for additional time post op helps? If there is any suggestions/guidance for the reader based on all the studies reviewed I would encourage you to include that in the Conclusion or make specific suggestions for future research.

Author Response

Obesity is certainly a challenge and as you point out not something that can be changed in an acute trauma situation.  It is difficult to know whether nonunion and failure of fixation was related to infection or metabolic (poor nutrition, low Vitamin D) reasons. I think providing suggestions in the Conclusion for the most significant findings would be important. If infection is the most significant complication is there any evidence that how we prep or redosing antibiotics for longer duration surgery or continuation of antibiotics for additional time post op helps? If there is any suggestions/guidance for the reader based on all the studies reviewed I would encourage you to include that in the Conclusion or make specific suggestions for future research.

Response:

We are grateful for the suggestion.  In relation to these complications, we have added to the conclusions:

In this patient group the index of suspicion of infection/non-union should be higher, with the threshold to be lower for starting anti-biotics and other interventions where appropriate. Post-operatively nutritional advice should be sought to address deficiencies in macro and micro dietary requirements, along with lifestyle support.  There is an urgent need for further research into strategic interventions in this patient group to improve outcomes and reduce complications.

Again, thank you for the suggestion on future research, we have added to the conclusion.

Reviewer 2 Report

Comments and Suggestions for Authors

Thank you for submitting this article which answers a certain important question. It shows the differences in the complications after orthopedic and trauma surgery in normal and overweight patients in the style of a systematic review.

I recommend checking the orthography again in the case of some errors at the beginning of the article.

Overall, it is not entirely clear whether there is already a review for this exact question. Furthermore, the different articles that were used in the table are not explained in their type and are not clearly rated. In my view, this leads to confusion for the reader. The findings as a whole are not new, but are well presented and well processed. 

Therefor I recommend the acceptance with major revisions. 

Author Response

1. Thank you for submitting this article which answers a certain important question. It shows the differences in the complications after orthopedic and trauma surgery in normal and overweight patients in the style of a systematic review.

Response:

Thank you.

2. I recommend checking the orthography again in the case of some errors at the beginning of the article.

Response:

We apologise if there were errors, we have endeavoured to correct these.

3. Overall, it is not entirely clear whether there is already a review for this exact question.

Furthermore, the different articles that were used in the table are not explained in their type and are not clearly rated. In my view, this leads to confusion for the reader.

Response:

There is not a systematic review/meta-analysis on the same topic.  The closest article on acute trauma surgery collected data up to 2012 (https://doi.org/10.1016/j.injury.2012.10.038). 

As can be seen from Table 1 there have been 18 studies since this date, which are included in our review.  Further, there have been other systematic reviews/meta-analysis but these related to elective surgeon rather than acute trauma surgery, which is the focus of our paper.

We have now described the studies and rated them, please see the tracked changes in Table 1.[1]

4. The findings as a whole are not new, but are well presented and well processed. 

Response:

Thank you.

5. Therefor I recommend the acceptance with major revisions. 

Response:

We are honoured to have our work published in the journal and we hope are revisions are sufficient.

[1]Guyatt GH, Oxman AD, Kunz R, Vist GE, Falck-Ytter Y, Schunemann HJ. What is “quality of evidence” and why is it important to clinicians? BMJ (Clinical research ed). 2008;336(7651):995-8 --- Guyatt GH, Oxman AD, Vist GE, Kunz R, Falck-Ytter Y, Alonso-Coello P, et al. GRADE: an emerging consensus on rating quality of evidence and strength of recommendations. BMJ (Clinical research ed). 2008;336(7650):924-6.------Guyatt G, Oxman AD, Akl EA, Kunz R, Vist G, Brozek J, et al. GRADE guidelines: 1. Introduction-GRADE evidence profiles and summary of findings tables. Journal of clinical epidemiology. 2011;64(4):383-94.

Round 2

Reviewer 2 Report

Comments and Suggestions for Authors

Thank you for revising the paper, for me these are sufficient.